# Vaccination Against Extracellular Vimentin Plus Doxorubicin for Canine Hemangiosarcoma

**DOI:** 10.3390/ijms26189096

**Published:** 2025-09-18

**Authors:** Diederik J. M. Engbersen, Lobke van Bergen, Emma N. Bos, Quinty Hansen, Arno Roos, Elisabeth J. M. Huijbers, Erica A. Faulhaber, Pancras C. W. Hogendoorn, Douglas H. Thamm, Arjan W. Griffioen

**Affiliations:** 1CimCure BV, 1066 CX Amsterdam, The Netherlands; 2Department of Pathology, Leiden University Medical Center, 2333 ZG Leiden, The Netherlands; 3IVC Animal Hospital Barendrecht, 2992 LC Barendrecht, The Netherlands; 4IVC Animal Hospital Nieuwegein, 3433 NP Nieuwegein, The Netherlands; 5Angiogenesis Laboratory, Department of Medical Oncology, Amsterdam University Medical Center, Cancer Center Amsterdam, 1081 HV Amsterdam, The Netherlands; 6Flint Animal Cancer Center, Department of Clinical Sciences, Colorado State University, Fort Collins, CO 80523, USA

**Keywords:** angiogenesis, sarcoma, angiosarcoma, hemangiosarcoma, immunotherapy, soft tissue tumour, vaccine, canine study

## Abstract

Angiosarcomas are highly aggressive soft tissue tumors with poor prognosis in both humans and dogs. In dogs, visceral hemangiosarcoma offers a relevant spontaneous model for evaluating novel therapies. Surgery alone yields a median survival of 1–3 months, and treatment with doxorubicin (DOX), alone or in combination (e.g., with propranolol), modestly extends median survival time to 5–7 months, with a 1-year survival of around 10%. We developed a conjugate vaccine technology, called immune Boost (iBoost), and hypothesized that combining DOX with an iBoost vaccine targeting extracellular vimentin (eVim) could improve survival without added toxicity. Twenty-three dogs with visceral hemangiosarcoma received six cycles of DOX every two weeks post-splenectomy, alongside four doses every other week of eVim iBoost immunotherapy, followed by maintenance vaccinations every two months. Outcomes were compared to historical controls treated with DOX alone. Compared to the control group the median overall survival time increased from 136 to 235 days (NS), restricted mean survival time at one year increased with 81 days (*p* = 0.02) and 1-year survival rate was 44% versus 14% (*p* = 0.0344). The combination was well-tolerated, with no systemic vaccine-related toxicity. Adding dog eVim vaccine to DOX appears to improve survival without added toxicity in dogs with hemangiosarcoma. These results support further clinical development, including evaluation in human angiosarcoma.

## 1. Introduction

Angiosarcoma (AS) represents a rare and aggressive subclass of soft tissue sarcoma, characterized by endothelial differentiation, typically seen in elderly patients [1,2]. A similar disease occurs in elderly dogs, where it is referred to in the veterinary literature as hemangiosarcoma (HSA). Canine HSA shares clinical, histopathological, and genetic features with human AS [3,4,5]. Both forms pose significant clinical challenges, as the effect of systemic treatment beyond surgery of resectable tumours is limited. Due to the rarity of AS, available treatment data are largely limited to retrospective case studies and there are currently no disease-specific evidence-based treatment guidelines beyond the general recommendations for soft-tissue sarcomas provided by the European Society for Medical Oncology (ESMO) and the National Comprehensive Cancer Network (NCCN) [1]. Some cases of AS arise following radiotherapy and are often associated with MYC gene amplification [6]. Retrospective studies suggest that paclitaxel may offer clinical benefit [7], although this has never been definitively confirmed in phase II-III trials.

For HSA, a more prevalent tumour in dogs, more extensive research has been conducted [8,9]. Adjuvant chemotherapy, typically with anthracyclines, such as doxorubicin (DOX) and epirubicin, can extend median overall survival time from 1–3 (with surgery alone) to 4–6 months [10,11,12]. However, 1-year survival is rarely achieved, and durable responses remain uncommon, as most dogs ultimately succumb to progressive metastatic disease.

Alternative therapies, including single agents or combinations with DOX, vincristine, cyclophosphamide, etoposide, carboplatin, dacarbazine and toceranib, have been explored but results were modest and inconsistent, offering no clear therapeutic advantage over single agents alone [13,14,15]. In the mid-1990s a study with liposome-encapsulated muramyl tripeptide phosphatidylethanolamine adjuvant immunotherapy resulted in significantly prolonged disease-free and overall survival in dogs with HSA [16]. The addition of propranolol, first reported by the group of Léauté-Labrèze in 2008 for the treatment of AS, has been receiving renewed interest [17]. However, no data are currently available regarding its efficacy in HSA [18,19,20]. More recently, three studies have evaluated combinations of surgery and DOX with adjunctive immunotherapies, including dendritic cell vaccination, non-specific immunotherapy with Immunocidin^®^ and peptide-based vaccination [21,22,23]. Among these, only the peptide-based approach demonstrated a potential improvement in overall survival.

Despite decades of research, treatment options for AS and HSA remain very limited, underscoring the need for new therapeutic targets and strategies [24,25,26,27]. A major barrier to effective treatment is the immunosuppressive tumor microenvironment, which compromises the effectiveness of conventional therapies [28,29]. Addressing this challenge requires novel approaches that not only target tumor cells, but also modulate the tumor vasculature, reshape the tissue microenvironment and engage the immune system.

A promising target that has recently gained attention in the context of immunotherapy is extracellular vimentin (eVim), which is secreted by tumour endothelial cells in the majority of solid tumours [30]. We developed a conjugate vaccine technology called immune Boost (iBoost) that is specifically designed to generate antibody responses to self-antigens [31]. Through this technology a vaccine was designed against eVim and subsequent research showed that eVim plays a crucial role in promoting both angiogenesis and immune evasion. Preclinical studies in mouse models of melanoma and colorectal cancer have shown that targeting eVim with monoclonal antibodies or vaccination inhibits tumour growth, highlighting its potential as an anti-angiogenic immunotherapy [30,32]. Building on these findings, a study in client-owned dogs with urothelial carcinoma showed that vaccination against vimentin was well tolerated, induced efficient antibody responses and doubled the median overall survival time, as compared to a historical control group [33].

In this study, we investigated the safety and therapeutic potential of combining the eVim iBoost vaccine with standard treatment using DOX, with or without propranolol, in dogs with spontaneous visceral HSA. We hypothesized that combining the vimentin-targeting vaccine with DOX would stimulate a robust anti-vimentin antibody response and produces improved survival outcomes over treatment with DOX alone.

## 2. Results

### 2.1. Study Population

Twenty-four dogs with histopathology-confirmed hemangiosarcoma (HSA, Figure 1A), 23 with splenic and 1 with cardiac HSA, and meeting the eligibility criteria for this study were enrolled from 2020 through 2024. One dog was excluded from analysis due to a protocol violation. Of the remaining 23 dogs, 15 received concomitant treatment with DOX and the dog-specific eVim vaccine. Three dogs received the eVim vaccine sequentially, immediately following completion of a doxorubicin (DOX) regimen. The remaining five dogs, whose owners declined chemotherapy, received dog eVim vaccine as monotherapy. In total, 14 dogs were administered propranolol in addition to their assigned treatment protocol. Baseline characteristics of the study population, along with those of the historical control group [14] are summarized in Table 1.

### 2.2. Immunohistochemical Analysis

Immunohistochemistry was performed to validate the expression of ERG (ETS-related gene), a marker used in the human clinic to confirm angiosarcoma. In concordance with observations in human AS, the dog tissues demonstrated specific and abundant overexpression of ERG in the vast majority of cancer cells (Figure 1A). Tumors were also found to be heavily infiltrated by leukocytes, as demonstrated in the CD45 staining (Figure 1A). HSA is a tumor of the vascular system and in dogs it was demonstrated to have a strong angiogenic phenotype [34]. Vimentin staining confirmed expression not only in the tumor cells but also prominently in the tumor-associated microvasculature (Figure 1A). As in many, if not all, solid tumors, and consistent with previous studies, vimentin was expected to be partially secreted and present in its extracellular form. These findings support the notion that extracellular vimentin (eVim) is a promising therapeutic target in canine HSA. Therefore, we evaluated the dog-specific iBoost eVim vaccine as a potential treatment for this disease.

### 2.3. Vaccination and Antibody Responses

A study was performed in dogs with HSA using the canine version of the eVim vaccine. Dogs enrolled in the study received a vaccination schedule consisting of four induction doses administered at two-week intervals (Figure 2A), followed by maintenance vaccinations given every eight weeks. All dogs developed detectable antibody responses against extracellular vimentin immediately after the first vaccination, with titers increasing progressively following the subsequent booster doses (Figure 2B). This robust immunogenicity demonstrates that the dog-specific iBoost eVim vaccine effectively induced an antibody-mediated immune response in all treated animals.

Notably, there was considerable inter-individual variability in antibody titers among the different dogs, a phenomenon commonly observed in outbred populations. The route of vaccine administration, i.e., subcutaneous (s.c.) versus intramuscular (i.m.) injections, did not significantly impact the magnitude of the antibody response, as comparable titers were seen in both groups (Figure 2C). Additionally, no marked differences in antibody responses were observed between male and female dogs (Figure 2D).

Antibody responses in dogs receiving eVim alone or sequentially after DOX were similar to those observed in the dogs receiving concomitant DOX and vaccine, suggesting that prior or concurrent chemotherapy did not impair the development of a vaccine-induced immune response.

These results confirm that the iBoost eVim vaccine is highly immunogenic in dogs with HSA, independent of sex, administration route, or chemotherapy status. The strong antibody responses across treatment groups further support investigation of the vaccine’s therapeutic potential in canine HSA.

### 2.4. Clinical Outcome

At the end of the follow-up period, one dog originally diagnosed with stage II HSA remained alive and progression-free, with a follow-up time of 1119 days. Of the remaining 22 dogs, 19 dogs were euthanised and three dogs died spontaneously, all due to suspected or confirmed progressive disease. One dog with stage I HSA developed liver metastases at day 128 but ultimately died because of a torsion of the mesentery at day 446, an event unrelated to tumor burden.

The overall median progression free interval (PFI) for the cohort was 142 days (range 24–1119) and median overall survival time (OS) for all clinical stages was 168 days (range 34–1119, Figure 3A). Survival analysis stratified by disease stage revealed a stage-dependent inverse correlation with outcome, with lower stages generally associated with longer survival (Figure 3B). Notably, the median OST for dogs with stage I and II was 235 days, which was nearly twice that of the historical control group (136 days, Table 2 and Figure 3D), but this difference was not significant (*p* = 0.18). It is noteworthy that the one dog with stage II cardiac HSA experienced a complete remission over a 2-month period of treatment (Figure 4).

Importantly, the 1-year survival significantly increased significantly (*p* = 0.0344) from 14% in the control group to 44% in the vaccinated study group (Table 2, Figure 3D). Restricted mean survival time (RMST) analysis at one year showed a significant advantage of 81 days (*p* = 0.02, Figure 3E). Among the 23 treated dogs, 14 received additional propranolol treatment. However, no additional improvement in survival or disease control was observed for this subgroup (Figure 3C). Interestingly, of the five dogs presenting with stage III disease, three showed clinical benefit. One dog had a complete response of its liver metastasis (see above), and two dogs experienced stable disease for 157 and 168 days. The remaining two stage III patients progressed despite treatment.

These findings suggest that the dog eVim vaccine, even in the context of advanced disease, can induce durable clinical benefit in a subset of patients and may significantly improve long-term survival outcomes in canine HSA.

### 2.5. Safety and Tolerability

The vaccination and combination treatment regimens were well-tolerated across the study population. Adverse events were limited to grade 1 and 2 toxicities, in all but one dog, primarily consisting of local injection site reactions, and mild transient vaccination-related clinical signs, such as lethargy and fatigue. These events were self-limiting and required no medical intervention in most cases. One dog showed grade 3 vomiting and grade 4 neutropenia which were attributed to concomitant DOX treatment rather than the vaccine itself (Table 3). Most adverse events occurred during the induction phase of the vaccination schedule. Two dogs were hospitalized, one dog for the abovementioned vomiting and one dog for a single day due to the owner’s concern over grade 2 adverse events, including lethargy, fatigue, and lameness in the limb near the injection site. This event also resolved without the need for further intervention. Injection site reactions were observed exclusively in dogs receiving subcutaneous injections; no such reactions occurred following intramuscular administration.

Overall, the safety profile of the iBoost eVim vaccine, both as monotherapy and in combination with DOX, was favorable and consistent with expectations for a therapeutic cancer vaccine and DOX treatment. The manageable and transient nature of the observed adverse events supports the continued clinical development of this approach in canine HSA.

## 3. Discussion

In this study, we tested a new cancer vaccine directed against extracellular vimentin (eVim) in client-owned dogs with visceral hemangiosarcoma (HSA). In this study we combined the vaccine with adjuvant doxorubicin (DOX) therapy after splenectomy. The data were compared to a historical control group receiving DOX only [14]. Since the control group exclusively involved stage I and II patients, the comparison could only be performed for those stages in the study group. The median OST (136 vs. 235 days) was increased, yet not significantly. Interestingly, we found that vaccination resulted in a significantly increased 1-year survival of 44%, as compared to 14% in the control group (*p* < 0.05). In the current study we also found a significant increase of the restricted mean survival time at one year with 81 days (*p* = 0.02) The combination treatment was well tolerated: only grade 1 and 2 injection site-related adverse events were seen. No systemic toxicity due to vaccine-induced antibody response. Importantly, the antibody response was not impacted by giving DOX simultaneously versus dogs receiving the vaccine only or after DOX. In this study, 3 dogs were included after having completed the DOX treatment. This may have introduced a selection bias, and these dogs did only later benefit from the vaccine treatment.

Angiosarcoma is a soft tissue sarcoma of endothelial origin that is rare in humans, but more commonly observed in dogs, where it is referred to as HSA. Despite differences in incidence, angiosarcomas in both species share significant genetic and histopathological similarities. In both humans and dogs, treatment options beyond surgical resection are limited and generally ineffective. In humans, there is no conclusive evidence supporting the benefit of adjuvant chemotherapy; it has been suggested that biologically targeted therapies, such as immune checkpoint inhibitors, and preferably anti-angiogenic agents may offer better options [35,36]. A similar situation exists in veterinary medicine, where no specific chemotherapy regimen or combination therapy has demonstrated clear superiority. The therapeutic benefit of systemic treatment remains modest, typically extending overall survival by only 3 months beyond surgery alone, for a total of approximately 4–6 months [9]. To date single-agent chemotherapy, most commonly DOX, remains the standard of care in dogs. However, DOX treatment is constrained by its cumulative dose limit of 180 mg/m^2^, which restricts the number of treatment cycles to a maximum of 5–6 doses, typically administered at 2–3-week intervals. This limited window of administration further reduces the potential for long-term disease control.

The vaccine used in this study has been designed to induce high antibody titers to the self-antigen extracellular vimentin (eVim). This was achieved using the conjugate vaccine technology immune Boost (iBoost) was specifically developed for this purpose [31]. The efficiency of the vaccine is high, as all dogs respond with antibody induction. Interestingly, the magnitude of the antibody response in different dogs varied significantly. This inter-individual variability was also noted in previous studies and in a recent study in dogs with urothelial carcinoma. In all these studies no consistent correlation has been observed between antibody titers and therapeutic efficacy in any of these settings. The mechanism-underlying antibody-mediated effects of the vaccine have not yet been fully elucidated, yet it can be expected that either functional blockade of the target and/or phenomena such as antibody-dependent cell cytotoxicity and complement activation are involved. Such diverse and possibly overlapping immune mechanisms, as well as expected variations in the polyclonality of the antibody response, may explain the lack of a direct, linear correlation between antibody titers and clinical outcomes.

A key outcome of this study was an observed improvement in median OST, which was nearly doubled compared to controls. However, this difference did not reach statistical significance. Notably, five dogs enrolled in the study received the vaccine as a monotherapy and therefore did not benefit from the additional survival advantage conferred by DOX treatment, estimated as an additional 136 days of survival. The median OS as presented in Figure 3D would likely have reached statistical significance if these dogs had received the combination therapy. Regardless, the Kaplan–Meier curves in Figure 3D and a significant increase in one-year survival (Table 2), indicating that the true therapeutic benefit becomes apparent in dogs that surpass the median survival threshold.

Recent studies have identified propranolol as a potential anti-cancer agent, including in sarcomas. This interest stems from observation made in infantile hemangioma, where treatment resulted in significant inhibition of hemangioma growth [37,38,39,40,41,42]. Based on these observations, we compared treatment with and without propranolol in this study. Propranolol inhibits beta-adrenergic signalling, which is known to stimulate tumor cell proliferation [43], angiogenesis [44] and metastasis formation [45]. Hence inhibition of this pathway can be expected to generate an anti-cancer activity. However, we did not observe any indication that adding propranolol to the vaccine plus DOX regimen conferred additional benefit. This may suggest that beta-adrenergic signaling plays a less prominent role in this tumor model than previously proposed, or that the conservative dosing at the lower end of the recommended dose range (0.5–1.5 mg/kg TID, [46] was used, or that the limited sample size in our study may have obscured potential effects.

The current has several other limitations. First, the study is an open-label single-arm prospective clinical study, that makes use of a historical control group for comparison. This may introduce temporal differences in the treatment of these dogs. However, adjuvant treatment and dose regimen with DOX remained unchanged as the care for dogs with hemangiosarcoma for decades, making it unlikely that temporal differences play a role. Second, next to the abovementioned group that received vaccination only, we also included a group of three dogs that received sequential vaccination after DOX. The latter group did not present a longer median OST (207 days). Therefore, we left this group in the analysis.

Taken together, these findings support further development of eVim-targeted vaccination strategies, particularly in combination with chemotherapy. Future studies should include larger, prospective cohorts to validate the survival benefit. Additionally, biomarker-driven patient stratification and integration of complementary immunomodulatory strategies, such as immune checkpoint blockade or agents targeting tumor vasculature, could further enhance therapeutic outcomes. Given the translational relevance of canine HSA to human angiosarcoma, these results also warrant exploration of this vaccine approach in human clinical settings.

## 4. Materials and Methods

### 4.1. Study Design and Participants

An open-label single-arm prospective clinical study in privately owned dogs was performed following approval by the Animal Ethics Committee of the VU University and the national Central Authority for Scientific Procedures on Animals (reg. no. CCDAVD11400202011305, 15 December 2020). The study was designed to assess the safety and efficacy of adjuvant dog eVim vaccine in combination with the maximal tolerated dose DOX in dogs with visceral HSA. Propranolol was allowed to be added to the adjuvant treatment per the treating veterinarian’s discretion. Owners who refused to have their dogs treated with chemotherapy were offered adjuvant dog eVim vaccine alone. Dogs with suspected HSA were completely staged with Thoracic radiographs, cardiac- and abdominal ultrasound, complete blood count and biochemistry profile before surgical intervention. Only dogs with histologically confirmed HSA by board certified veterinary pathologists were eligible. Dogs with suspected cardiac disorders, such as dilated cardiomyopathy, or recent (two weeks before first vaccination) or current treatment with immunosuppressive therapy for other diseases and/or prior malignancies were excluded per the principal investigator’s discretion. Upon written owner consent, dogs were included in the study; these dogs were not hospitalised and lived at home with their owners. After the initial vaccination, the dogs received three booster vaccinations at 2-week intervals, followed by maintenance vaccinations at 2-month intervals for the duration of the study. Initially, vaccinations were given subcutaneously (s.c., n = 12) in the groin. Feedback from the previous study in urothelial carcinoma indicated that an intramuscular (i.m.) vaccination would be preferred over the s.c. one for two reasons: first owners typically worried about the induration/ulceration seen in up to 50% of the dogs at the injection site and secondly veterinarians considered an i.m. vaccination a more patient friendly way of application. An animal safety study was performed and showed that the i.m. vaccination generated robust antibody titers and was safe (unpublished results). Subsequently, the route of administration was switched to i.m. (n = 11) in the upper leg.

The dog eVim vaccine was composed of 500 μg recombinant fusion protein TRXtr-dogVimentin (TRXtr-dVim) (canis lupus familiaris, NCBI ref seq NM_001287023.1) adjuvanted with 375 μg phosphorothioate stabilised CpG 2006 oligonucleotide (5′-TCG-TCG-TTT-TGT-CGT-TTT-GTC-GTT-3′; Eurogentec, Fremont, CA, USA) and 10% Montanide Gel 01PR (36067D, Seppic, Paris, France, final concentration Montanide gel 5%). Maintenance vaccinations were given without CpG. The vaccine was dosed according to body weight, with dogs > 25 kg receiving the full dose of 500 µg, dogs 10–25 kg half the dose and dogs < 10 kg one third of the initial dose.

The DOX treatment regimen consisted of 6 doses of 30 mg/m^2^ in dogs > 10 kg and 25 mg/m^2^ in dogs < 10 kg, given at 2-week intervals (Figure 1A). During the study, there was a temporary shortage of DOX and 5 out of 18 dogs received as dose 5 and 6 Epirubicin instead of DOX at the same dose as for DOX. All but one dog received all 6 doses, one dogs received 5 doses of DOX. Dogs that received propranolol were administered 0.5 mg/kg every 8 h for the duration of the study.

### 4.2. Immune Response Measurement

Tissue morphology of HSA tissues was visualised by hematoxylin and eosin (HE) staining. immunohistochemistry (IHC) was used to evaluate ERG, CD45, and vimentin expression. Protocols were used as described before [33,47]. Blood samples were taken at the start of the study, before vaccination, and at each follow-up visit. Blood was stored overnight at 4 °C to coagulate, and the next day the samples were centrifuged at 4650 rcf, 10 min in a microcentrifuge to collect the serum. Serum samples were stored at −20 °C. Indirect ELISA was performed to determine total dogVim antibody levels as previously described [33].

### 4.3. Anti-eVim Antibody ELISA

Antibody titers were measured in duplicate in 96-well ELISA plates (F96 Maxisorp, Nunc A/S, Roskilde, Denmark) coated with 4 µg/mL recombinant dog vimentin protein and subsequently blocked with 4% milk/PBS (100 µL/well) (sc-2325, Santa Cruz Biotechnology, Dallas, TX, USA), both for 1 h at 37 °C. Plates were incubated with biotinylated polyclonal goat anti-canine IgG (6070-08, Southern Biotech, Birmingham, AL, USA, 0.25 µg/mL) for 45 min at 37 °C and streptavidin-horseradish peroxidase (P0397, Dako Cytomation, Carpinteria, CA, USA, 0.45 mg/mL) for 30 min. Plates were developed and measured as previously published [33].

### 4.4. Patient Monitoring and Evaluation

Dogs were re-assessed at each visit according to the treatment schedule (Figure 2A). At each visit, a physical examination, cardiac and abdominal ultrasound, blood count and a biochemistry panel (containing urea, creatinine, ALT, ALP, total protein, albumin, globulin and glucose) were performed. Thoracic radiographs were performed every 6 months or indicated by clinical respiratory symptoms. The dogs were evaluated by the veterinarian for clinical response according to the VCOG consensus document on response evaluation criteria for solid tumours in dogs v1.0 based on physical condition, abdominal ultrasound and X-ray of the thorax [48]. Adverse events were assessed according to the Veterinary Cooperative Oncology Group-Common Terminology Criteria for Adverse Events v2 [49].

### 4.5. Statistical Analysis

Means of antibody titers were compared with a Mann–Whitney U test or a two-tailed Student’s *t* test for calculation of significance. Kaplan–Meier curves were made for overall survival, and a Gehan-Breslow-Wilcoxon test was applied. 1-year OST was analysed by one-sided Fisher’s Exact test and RMST analyses. GraphPad Prism^®^ v10 was used for the analyses, except for RMST, where SurvRM2 v1.0-4. The level of significance was set to *p* < 0.05.

## Figures and Tables

**Figure 1 ijms-26-09096-f001:**
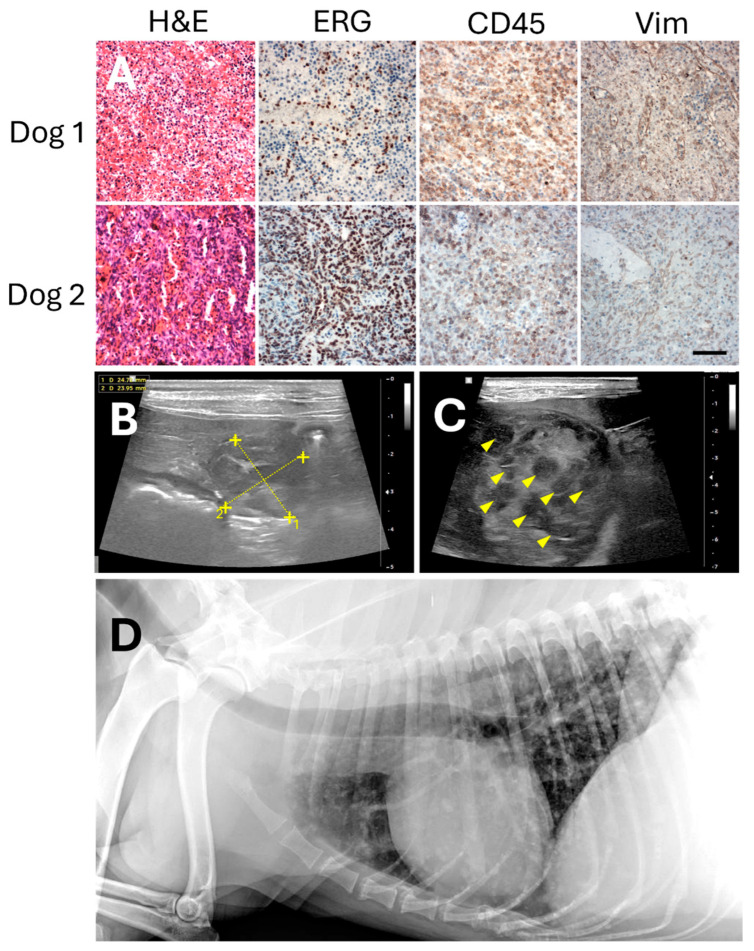
Immunohistochemistry, ultrasound and CT of canine HSA. (**A**) Representative images of HSA of two different dogs of Hematoxylin/eosin (HE) staining and staining for ETS-related Gene (ERG), CD45 and vimentin are shown. Scale bar in lower right panel represents 100 μm. (**B**) Abdominal ultrasound image of a dog with a primary splenic HSA. Yellow cross indicates the tumor dimensions. (**C**) Ultrasound image of the same dog showing liver metastases developed on day 276 post-surgery, indicated by yellow arrowheads. (**D**) Thoracic radiograph of a dog with splenic HSA, showing abundant lung metastases.

**Figure 2 ijms-26-09096-f002:**
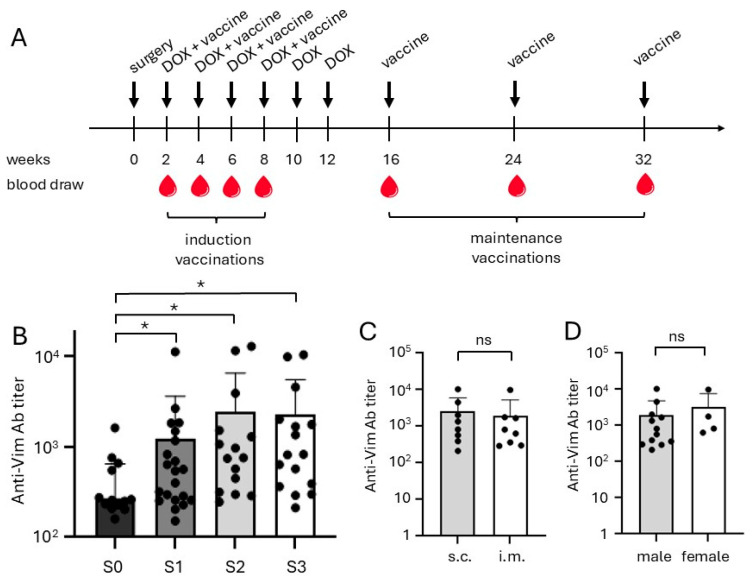
Vaccination schedule and antibody responses in dogs vaccinated with dog eVim vaccine. (**A**) Schematic overview of the study set-up. Thoracic X-ray, cardiac and abdominal ultrasound, blood count and biochemistry profiles were performed prior to surgical intervention. Diagnosis was made post-surgery after histopathological examination. Induction vaccinations were given at 2-week intervals and maintenance vaccinations every 2 months. During the induction phase, blood samples were taken every two weeks (S0–S3) at the time point of vaccination and during maintenance (S4 and up). (**B**) Anti-eVim antibody titers at start of treatment (S0) and after subsequent vaccinations. Titers at S1, S2 and S3 are significantly different from S0 (*p* = 0.039, 0.0321 and 0.0177, respectively). (**C**) Anti-eVim antibody titers divided by vaccination route (s.c. and i.m.). No significant differences observed (ns). (**D**) Anti-eVim antibody titers divided by sex (male, female) at time point S3. No significant differences observed (ns). Data are depicted as mean ± SEM. * *p* < 0.05.

**Figure 3 ijms-26-09096-f003:**
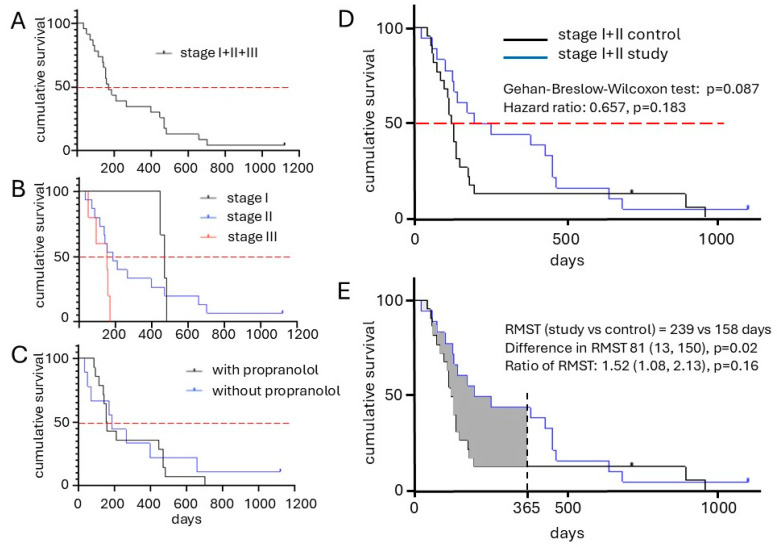
Clinical response in dogs vaccinated with dog eVim vaccine. (**A**) Kaplan–Meier curve of probability (%) of OS of the total study group. (**B**) Kaplan–Meier curve of probability (%) of OS of the study group by HSA clinical stage (stage I—black line; stage II—blue line; stage III—red line). (**C**) Kaplan–Meier curve of probability (%) of OS of the study group by additional treatment with propranolol (no propranolol—black line; with propranolol—blue line). (**D**) Kaplan–Meier curve of probability (%) of OS of clinical stage I and II population of the study group (stage I + II study—blue line) compared the historical control group (stage I + II control—black line) published before [23]. The red dotted lines mark the median overall survival time. (**E**) Restricted mean survival time analysis. A significant 81 days difference was observed (*p* = 0.02).

**Figure 4 ijms-26-09096-f004:**
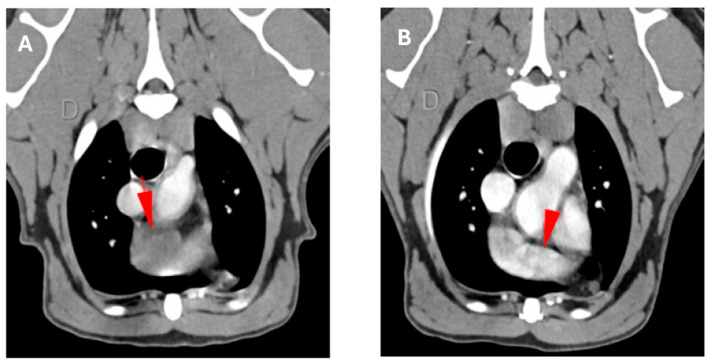
Radiological response by CT of a complete remission. (**A**) CT scan of a dog with cardiac HSA showing a right auricular expansion lesion of 2.7 cm (red arrow) with pericardial effusion (seen on 11 July 2022). (**B**) CT scan of the same dog showing no evidence of the right auricular expansion lesion (red arrow) and no pericardial effusion (regular aspect, seen on 20 October 2022). D = dexter (right side).

**Table 1 ijms-26-09096-t001:** Study population characteristics.

Dogs (n)		Gender (n)				Breed * (n)		Weight (kg)	Age (yr)	Clinical Stage (n)		
		Female		Male		Single	Mixed	Median	Mean	I	II	II
		Intact	Spayed	Intact	Castrated							
		%		%		%	%	Range	Range	%	%	%
Study	23	0	7	9	7	15	8	28.9	9.2	3	15	5
		0%	30%	39%	30%	65%	35%	5.2–64.0	1.9–13.6	13%	65%	22%
Control	22	0	11	2	9	15	7	25	N.R.	7	15	0
		0%	50%	9%	41%	68%	32%	8.4–42.0	N.R.	32%	68%	0%

* Dogs were either single breed or a mix of breeds. Study: Jack Russell Terrier, Cocker Spaniel, Dachshund, Viszla, French Bulldog (2), Riessen Schnautzer, Nova Scotia Duck Tolling Retriever, Labrador Retriever (2), Beagle, German Shepherd, Sheltie, Bull Mastiff, Frisian Stabyhoun. Control: Golden Retriever (4), German Shepherd (3), Schnauzer (2), Airdale, Bernese Mountain, Scottish Terrier, Poodle, Border Collie, Labrador Retriever.

**Table 2 ijms-26-09096-t002:** Overall survival in dogs with visceral HSA.

Overall Survival Time (d)	Study				Control				Significance ***	
	n	Median	Range	1-yr OS *	n	Median	Range	1-yr OS	*p*-Value **	Y/N
Stage I	3	469	446–348	100%	7	139	70–911	14%	0.0333	Y
Stage II	15	186	34–235	33%	15	128	54–975	13%	0.1949	N
Stage I + II	18	235	34–1119	44%	22	136	54–975	14%	0.0344	Y
Stage III	5	153	34–168	0%						
Stage I–III	23	168	24–1119	35%						

* OS: Overall Survival, **: *p*-value: one-sided Fisher’s Exact test, *** Significance: *p* < 0.05.

**Table 3 ijms-26-09096-t003:** Treatment-related adverse events.

Treatment Related Adverse Events	Events/Dogs (n = 23)							
(n)	Grade 1		Grade 2		Grade 3		Grade 4	
	s.c. * (n = 12)	i.m. ** (n = 11)	s.c. (n = 12)	i.m. (n = 11)	s.c. (n = 12)	i.m. (n = 11)	s.c. (n = 12)	i.m. (n = 11)
Injection side adverse events								
-Injection site reactions	8/4	0/0	3/2	1/1	0/0	0/0	0/0	0/0
-Lameness local extremity	0/0	4/2	2/2	0/0	0/0	0/0	0/0	0/0
Systemic adverse events								
-Diarrhoea	0/0	0/0	0/0	0/0	0/0	1/1	0/0	0/0
-Lethargy/fatigue/performance	1/1	0/0	0/1	0/0	0/0	0/0	0/0	0/0
-Neutropenia	0/0	0/0	0/0	0/0	0/0	0/0	0/0	1/1
-Vomiting	0/0	1/1	0/0	1/1	0/0	0/0	0/0	0/0
**Total adverse events**	**9/6**	**5/3**	**6/5**	**2/2**	**0/0**	**1/1**	**0/0**	**1/1**

* s.c.: sub cutaneous vaccination; ** i.m.: intra muscular vaccination.

## Data Availability

Data is contained within the article.

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
