# Peer review of "Vaccination Against Extracellular Vimentin Plus Doxorubicin for Canine Hemangiosarcoma"

_ijms, 2025, doi:10.3390/ijms26189096_

Round 1

Reviewer 1 Report

Comments and Suggestions for Authors

Dear Authors,

all data and in general - manuscript are well written and very interesting.

I believe that standardisation of nomenclature is needed - angiosarcoma vel haemangiosarcoma. It is a little bit confusing and may be misleading.

The layout is incorrect – the introduction should be followed by materials and methods.

I also sugest to change the figure 1 - the photographs of vimentin and CD45 are blurry.

Author Response

Dear Authors,

all data and in general - manuscript are well written and very interesting.

I believe that standardisation of nomenclature is needed - angiosarcoma vel haemangiosarcoma. It is a little bit confusing and may be misleading.

Answer: We do see the point on the nomenclature. However in the human situation hemangiosarcoma does not exist or is very rare, but angiosarcoma is a disease that does exist in humans. When we refer to human disease we use “angiosarcoma”, when we refer to canine disease we use hemangiosarcoma. It has been introduced in the text of the introduction section - line 60-63

The layout is incorrect – the introduction should be followed by materials and methods.

Answer: We have put the M&M section in the order as presented in the Instructions for Authors.

I also sugest to change the figure 1 - the photographs of vimentin and CD45 are blurry.

Answer: Correct, the quality of the figure got lost in transition. We will upload a hi-res.

Reviewer 2 Report

Comments and Suggestions for Authors

The study investigates the combination of an extracellular vimentin–targeted vaccine (iBoost) with doxorubicin after splenectomy in dogs with hemangiosarcoma. The results are promising, showing good tolerability and a potential survival benefit compared with historical controls. However, significant methodological limitations reduce the robustness of the conclusions. Further refinement and clarification are needed before the findings can be considered for publication.

  1. Non-randomized design – The use of historical controls instead of a concurrent randomized control arm introduces selection and temporal biases.
  2. Small and heterogeneous sample – Only 23 dogs were included, with varied clinical stages (I–III) and treatment timing (concurrent vs. sequential vaccination).
  3. Treatment variability – Some dogs received propranolol or substituted epirubicin for doxorubicin, confounding attribution of observed effects to the vaccine.
  4. Mixed population – Three dogs began vaccination after completing chemotherapy, potentially biasing survival analyses.
  5. Limited mechanistic insight – Immunologic evaluation was restricted to anti-eVim antibody titers and basic IHC markers; deeper immune profiling is needed.
  6. Uncertain outcome classification – Some deaths lacked necropsy confirmation, risking endpoint misclassification.
  7. Efficacy inconsistency – While one-year survival improved significantly, the median OS difference was not statistically significant.
  8. Administration differences – Adverse events varied between subcutaneous and intramuscular injection, suggesting the need for delivery standardization.

Minor Comments:

  1. Clarify criteria for including stage III cases and justify their combination with early-stage disease in analyses.
  2. Provide more detail on dose timing and intervals for concurrent vs. sequential vaccination groups.
  3. Include exact P-values for all key comparisons in the results.

Recommendation for Authors:

The manuscript addresses an important and underexplored therapeutic approach in veterinary oncology. However, the limitations in study design, sample size, and treatment uniformity should be clearly acknowledged in the discussion. Additional details on methodology and more robust immune response data would strengthen the manuscript.

Author Response

Open Review #2

Comments and Suggestions for Authors

The study investigates the combination of an extracellular vimentin–targeted vaccine (iBoost) with doxorubicin after splenectomy in dogs with hemangiosarcoma. The results are promising, showing good tolerability and a potential survival benefit compared with historical controls. However, significant methodological limitations reduce the robustness of the conclusions. Further refinement and clarification are needed before the findings can be considered for publication.

  1. Non-randomized design – The use of historical controls instead of a concurrent randomized control arm introduces selection and temporal biases.

Answer: we do agree with the limitations of the presented study. Indeed instead of an randomized control arm we made use of a historical control arm. For our study we enrolled all dogs with histopathologically confirmed hemangiosarcoma, which precludes selection of patients. Of course there might a temporal bias. This has been explained in a new section in the discussion, addressing all limitations of the current study. Line 381-389

  1. Small and heterogeneous sample – Only 23 dogs were included, with varied clinical stages (I–III) and treatment timing (concurrent vs. sequential vaccination).

Answer: We acknowledge the limitations of the sample composition. We have presented the relevant results of the OS per stage (I-III) in figure 3B. The OS survival results of the dogs receiving sequential vaccinations have not presented separately, as this group consisted of only three dogs. We agree with the reviewer that this result may be presented as well. Therefore, we have mentioned this result now in the revised version – line 386-389

  1. Treatment variability – Some dogs received propranolol or substituted epirubicin for doxorubicin, confounding attribution of observed effects to the vaccine.

Answer: This point is well taken. The veterinarians were allowed to add propranolol to dogs in the study in case it was part of their standard treatment with DOX. This was the case for 14 out of the 23 dogs. The results of these 2 groups have been analyzed separately, we did not observe any positive nor negative effect of the propranolol as we have presented in Figure 3C and has been addressed in the discussion. As noted by the reviewer some dogs did receive epirubicin instead of doxorubicin, due to temporary market shortage. The manuscript has now been revised as to for which doses (dose 5 and 6 of the DOX regimen) and how many, in 5 out of 17 dogs, this was the case. Epirubicin is considered to be as effective as doxorubicin, yet has been less studied (Kim et al. 2007). - Line 440-442

  1. Mixed population – Three dogs began vaccination after completing chemotherapy, potentially biasing survival analyses.

Answer: This issue was already mentioned in point 2. The three dogs receiving sequential treatment did not show any markedly different survival. In addition, excluding them does not change the median overall survival of 235 days. There we did not exclude these dogs from the final analyses. We addressed this issue in a new paragraph of the revised discussion section.

  1. Limited mechanistic insight – Immunologic evaluation was restricted to anti-eVim antibody titers and basic IHC markers; deeper immune profiling is needed.

Answer: (Partial) cystectomy is not part of the standard treatment of canine UC, nor was necropsy part of the protocol. Consequently no tumor tissue was available for further immune profiling.

  1. Uncertain outcome classification – Some deaths lacked necropsy confirmation, risking endpoint misclassification.

Answer: The reviewer is right on this issue. However, necropsy was not part of the protocol and for that reason we were not able to perform necropsy confirmation. Therefore, 21 out 23 dogs have been classified to die from suspected or confirmed progressive disease. We do this because of the very aggressive nature of the disease. One dog died spontaneously because of a torsion of the mesentery, an event unrelated to tumor burden and one dog was alive still at the end of follow-up. The relevant result section has been revised for more clarity. – line 226-227

  1. Efficacy inconsistency – While one-year survival improved significantly, the median OS difference was not statistically significant.

Answer: The reviewer is right: overall survival was longer, but not significantly. The main result of the study is the enhanced 1-year survival. The revised manuscript now also contains data on restricted mean survival time analysis (RMST), see the suggestion by reviewer 3, which also present significant difference at 1-year. Line: 236, 239-242

  1. Administration differences – Adverse events varied between subcutaneous and intramuscular injection, suggesting the need for delivery standardization.

Answer: This is true. Adverse events were exclusively seen at the injection site. From another study we learned that injection site toxicity is much less in IM as compared to SC administration. Therefore we switched in this study to IM application. As we did not see any difference in antibody responses (see Figure 2C), we favor the view that this switch did not impact the study.

Minor Comments:

  1. Clarify criteria for including stage III cases and justify their combination with early-stage disease in analyses.

Answer: The control group consisted only of stage I and II cases, therefore we also limited OS analysis that those groups. Yet we treated five dogs with stage III disease. These were analyzed as a separate group (see Figure 3B). We think this is also interesting information and sufficiently clear in the M&M section.

  1. Provide more detail on dose timing and intervals for concurrent vs. sequential vaccination groups.

Answer: This issue is well taken. These three dogs received vaccination immediately following completion of the DOX treatment. We have added these details in the revised results section. - line 126

  1. Include exact P-values for all key comparisons in the results.

Answer: This issue is well taken, we have added P-values for all key comparisons

Recommendation for Authors:

The manuscript addresses an important and underexplored therapeutic approach in veterinary oncology. However, the limitations in study design, sample size, and treatment uniformity should be clearly acknowledged in the discussion. Additional details on methodology and more robust immune response data would strengthen the manuscript.

Answer: this is a good idea indeed. We added an extra section in the Discussion on all limitations of the discussion. – line 381-389

Reviewer 3 Report

Comments and Suggestions for Authors

This manuscript presents a single-arm prospective study of an extracellular vimentin vaccine (iBoost platform) combined with doxorubicin in client-owned dogs with visceral hemangiosarcoma. The work is interesting and of potential translational significance, but in its current form there are several substantive issues that need to be addressed before it can be considered for publication.

Major scientific and methodological concerns

  1. Comparator and confounding. The principal comparison is against a historical doxorubicin cohort. This introduces unavoidable biases: differences in staging, supportive care, imaging intervals, and owner decision-making may confound outcomes. In addition, within the study cohort there was heterogeneity: some dogs received propranolol, some received epirubicin substitution, and five received vaccine monotherapy. These variations obscure attribution of benefit. At a minimum, sensitivity analyses should be provided excluding monotherapy, excluding epirubicin substitutions, and stratifying by propranolol use.

  2. Primary endpoint. Median overall survival was numerically longer (235 versus 136 days) but not statistically significant, while one-year survival was significantly improved. The manuscript must pre-specify the primary endpoint and provide proper statistical analysis, including hazard ratios with confidence intervals, restricted mean survival times, and 95% confidence intervals for one-year survival proportions. Without this, the results remain exploratory.

  3. Handling of propranolol. The null result for propranolol should be interpreted with caution. Possible explanations include small sample size, low dose, or tumor biology. This should be discussed in more detail, and the actual doses administered and adherence reported.

  4. Immunogenicity. While robust anti-eVim titers were observed in all dogs, there is no exploration of correlation between antibody response and outcome. Even if no linear correlation is present, scatterplots, regression analysis, and landmarked Cox models would be informative. The inclusion of isotype data or functional assays would add strength to the mechanistic claims.

  5. Pathology. ERG staining is appropriate for confirming endothelial differentiation, but vimentin IHC only confirms expression. Claims about extracellular vimentin should be supported by more direct evidence. Inclusion of CD31 or von Willebrand factor, as well as extracellular localization, would strengthen the pathology section.

  6. Safety. Adverse events are well characterized using VCOG criteria. However, the manuscript should specifically comment on wound healing and vascular recovery, given the target is extracellular vimentin.

  7. Translational context. The discussion correctly highlights similarities between canine hemangiosarcoma and human angiosarcoma. However, prior immune-based interventions in canine hemangiosarcoma (such as muramyl tripeptide adjuvant therapy) should be more fully acknowledged as precedent.

Reporting and presentation issues

  1. The manuscript header contains inconsistencies: the year is listed as 2021, yet the enrollment extends to 2024 and references to 2025 are included. The DOI is still a placeholder. These must be corrected for scholarly integrity.

  2. Author information is duplicated in places (for example, “Correspondence: Correspondence”). ORCID numbers must be properly formatted.

  3. Tables require clarification. Table 1 has unclear headers (“Single/Mixed”) and should specify what these categories mean. Table 2 should present confidence intervals for survival estimates. Kaplan–Meier curves should display numbers at risk.

  4. Figures should include scale bars, magnifications, and clear labeling. CT images should show dates or time intervals.

  5. The abstract should be tightened to reflect the design limitations, specifying that comparison is against historical controls and that the survival improvement did not reach statistical significance for median overall survival.

Author Response

Open Review #3

Comments and Suggestions for Authors

This manuscript presents a single-arm prospective study of an extracellular vimentin vaccine (iBoost platform) combined with doxorubicin in client-owned dogs with visceral hemangiosarcoma. The work is interesting and of potential translational significance, but in its current form there are several substantive issues that need to be addressed before it can be considered for publication.

Major scientific and methodological concerns

  1. Comparator and confounding. The principal comparison is against a historical doxorubicin cohort. This introduces unavoidable biases: differences in staging, supportive care, imaging intervals, and owner decision-making may confound outcomes. In addition, within the study cohort there was heterogeneity: some dogs received propranolol, some received epirubicin substitution, and five received vaccine monotherapy. These variations obscure attribution of benefit. At a minimum, sensitivity analyses should be provided excluding monotherapy, excluding epirubicin substitutions, and stratifying by propranolol use.

Answer: We understand this issue of the reviewer. We addressed all these issues in response to the critiques by reviewer 2. Indeed the study has some relevant limitations. The issues concerning epirubicin treatment as a replacement for DOX, as well the additional administration of propranolol in a subset of dogs has been addressed in the response to reviewer number 2. The study contained five dogs that only received the vaccine. Although excluding these would likely have improved the OS data, we decided to leave these dogs in to raise the power of the analysis. This has been addressed in the revised discussion section.

  1. Primary endpoint. Median overall survival was numerically longer (235 versus 136 days) but not statistically significant, while one-year survival was significantly improved. The manuscript must pre-specify the primary endpoint and provide proper statistical analysis, including hazard ratios with confidence intervals, restricted mean survival times, and 95% confidence intervals for one-year survival proportions. Without this, the results remain exploratory.

Answer: The primary endpoints have been listed now in the revised version of the M&M section. We added data on confidence intervals, hazard ratios and restricted mean survival times . Line 444-445. And we added additional statistical details where relevant and added a new panel in Figure 3 (E) to the results section with the outcome of the RMST analysis.

  1. Handling of propranolol. The null result for propranolol should be interpreted with caution. Possible explanations include small sample size, low dose, or tumor biology. This should be discussed in more detail, and the actual doses administered and adherence reported.

Answer: This is an important point. As stated above we allowed the use of propranolol. When stratified for this extra treatment we did not see differences in OS. Indeed the study was not dedicated for investigating the combination treatment. We also understand that the propranolol dose we used was rather low and no formal documentation of dose adherence of propranolol was part of the protocol. We have no intentions to further test this combination. Nevertheless, the lack of an effect on OS made us decide that this subgroup can safely be kept in the overall analysis.

  1. Immunogenicity. While robust anti-eVim titers were observed in all dogs, there is no exploration of correlation between antibody response and outcome. Even if no linear correlation is present, scatterplots, regression analysis, and landmarked Cox models would be informative. The inclusion of isotype data or functional assays would add strength to the mechanistic claims.

This is indeed interesting. However, in numerous preclinical models we never saw a clear correlation between level of antibody response and anti-tumor effect. Also in a previous study with this vaccine in dogs with urothelial carcinoma no such effect was observed. We do not think that adding correlation plots is informative, nevertheless, this is addressed in the discussion on this issue. Page, line

  1. Pathology. ERG staining is appropriate for confirming endothelial differentiation, but vimentin IHC only confirms expression. Claims about extracellular vimentin should be supported by more direct evidence. Inclusion of CD31 or von Willebrand factor, as well as extracellular localization, would strengthen the pathology section.

This is correct. Indeed, the vimentin staining was only performed to show that vimentin is overexpressed in HSA tissue. The extracellular feature is very difficult to show. We did that in previous studies (Van Beijnum et al, Nat. Comm 2022) and we assumed that this feature is also relevant in other cancers. The fact that we see a survival benefit confirms this. A specific staining for vasculature would have been informative, however, since the tumor cells are of endothelial origin this was considered difficult. Notwithstanding this issue, the visualization of microvessels was performed by staining for vimentin (see Figure 1A).

  1. Safety. Adverse events are well characterized using VCOG criteria. However, the manuscript should specifically comment on wound healing and vascular recovery, given the target is extracellular vimentin.

This has been shown in previous research. Vaccination against extracellular vimentin is safe, as was published before (Van Beijnum et al, Nature Comm, 2022). It was shown in studies keeping mice long-term hyperimmune and in wound healing studies in mice. Also in a previous canine study, the vaccination was shown not to affect recovery after surgery (Engbersen et al, Cancers, 2023).

  1. Translational context. The discussion correctly highlights similarities between canine hemangiosarcoma and human angiosarcoma. However, prior immune-based interventions in canine hemangiosarcoma (such as muramyl tripeptide adjuvant therapy) should be more fully acknowledged as precedent.

Answer: Well taken, we have addressed this in the original submission, but we added the correct reference to this work. – Line 84 (Ref 16)

Reporting and presentation issues

  1. The manuscript header contains inconsistencies: the year is listed as 2021, yet the enrollment extends to 2024 and references to 2025 are included. The DOI is still a placeholder. These must be corrected for scholarly integrity.

True. We suppose that this will be corrected in the publication process.

  1. Author information is duplicated in places (for example, “Correspondence: Correspondence”). ORCID numbers must be properly formatted.

We saw this too. It was the result from formatting the manuscript for IJMS layout.

  1. Tables require clarification. Table 1 has unclear headers (“Single/Mixed”) and should specify what these categories mean. Table 2 should present confidence intervals for survival estimates. Kaplan–Meier curves should display numbers at risk.

This has been corrected and noted in the legend of the figures and tables

  1. Figures should include scale bars, magnifications, and clear labeling. CT images should show dates or time intervals.

This has been corrected. Scale bars is noted in the legend of Figure 1 and the dates of the CT images have been noted in the legend of Figure 4.

  1. The abstract should be tightened to reflect the design limitations, specifying that comparison is against historical controls and that the survival improvement did not reach statistical significance for median overall survival.

This has been done. – Line 48-49

Round 2

Reviewer 3 Report

Comments and Suggestions for Authors

No further improvement needed.